# The Molecular Networks of microRNAs and Their Targets in the Drug Resistance of Colon Carcinoma

**DOI:** 10.3390/cancers13174355

**Published:** 2021-08-28

**Authors:** Francesca Crudele, Nicoletta Bianchi, Annalisa Astolfi, Silvia Grassilli, Federica Brugnoli, Anna Terrazzan, Valeria Bertagnolo, Massimo Negrini, Antonio Frassoldati, Stefano Volinia

**Affiliations:** 1Department of Translational Medicine, University of Ferrara, Via Luigi Borsari 46, 44121 Ferrara, Italy; francesca.crudele@unife.it (F.C.); nicoletta.bianchi@unife.it (N.B.); annalisa.astolfi@unife.it (A.A.); silvia.grassilli@unife.it (S.G.); federica.brugnoli@unife.it (F.B.); anna.terrazzan@edu.unife.it (A.T.); valeria.bertagnolo@unife.it (V.B.); m.negrini@unife.it (M.N.); 2Laboratory for Advanced Therapy Technologies (LTTA), Via Fossato di Mortara 70, 44121 Ferrara, Italy; 3Department of Oncology, Azienda Ospedaliero-Universitaria St. Anna di Ferrara, Via A. Moro 8, 44124 Ferrara, Italy; antonio.frassoldati@unife.it

**Keywords:** miRNA, 5-fluorouracil, oxaliplatin, doxorubicin, cisplatin, irinotecan, colon cancer, non-coding RNA

## Abstract

**Simple Summary:**

We systematically reviewed the recent scientific publications describing the role of microRNAs in the regulation of drug resistance in colon cancer. To clarify the intricate web of resulting genetic and biochemical interactions, we used a machine learning approach aimed at creating: (i) networks of validated miRNA/target interactions involved in drug resistances and (ii) drug-centric networks, from which we identified the major clusters of proteins affected by drugs used in the treatment of colon cancer. Finally, to facilitate a high-level interpretation of these molecular interactions, we determined the cellular pathways related with drug resistance and regulated by the miRNAs in colon cancer.

**Abstract:**

Drug resistance is one of the major forces driving a poor prognosis during the treatment and progression of human colon carcinomas. The molecular mechanisms that regulate the diverse processes underlying drug resistance are still under debate. MicroRNAs (miRNAs) are a subgroup of non-coding RNAs increasingly found to be associated with the regulation of tumorigenesis and drug resistance. We performed a systematic review of the articles concerning miRNAs and drug resistance in human colon cancer published from 2013 onwards in journals with an impact factor of 5 or higher. First, we built a network with the most studied miRNAs and targets (as nodes) while the drug resistance/s are indicated by the connections (edges); then, we discussed the most relevant miRNA/targets interactions regulated by drugs according to the network topology and statistics. Finally, we considered the drugs as nodes in the network, to allow an alternative point of view that could flow through the treatment options and the associated molecular pathways. A small number of microRNAs and proteins appeared as critically involved in the most common drugs used for the treatment of patients with colon cancer. In particular, the family of miR-200, miR34a, miR-155 and miR-17 appear as the most relevant microRNAs. Thus, regulating these miRNAs could be useful for interfering with some drug resistance mechanisms in colorectal carcinoma.

## 1. The Curated Networks of MiRNAs and Their Targets in Colon Cancer Drug Resistance

In our previous works, we dissected the relations between long non-coding RNAs (lncRNAs), or microRNAs (miRNAs), and drug resistance in various types of carcinomas [1]; successively, we focused on non-coding RNAs and their targets in breast cancer [2]. Here, we merged these two approaches to systematically review the recent literature. Overall, our effort was aimed at the identification of the crucial central miRNAs and their targets in the pathways involved in the drug resistance of colon carcinoma. We restricted our study to 499 research articles listed in PubMed-NCBI and published after 2012 (Appendix A). The query we used for selection of the manuscripts on microRNA and drug resistance in colon cancer is reported in the Appendix A. Among those, we selected 102 research articles (not reviews or metagenomics studies) based solely on the journal impact factor (at least 5.0). We preferred the impact factor rather than the number of citations, since the latter is largely influenced by the publication age and might not be a fair criterium for papers published recently. Then, we carried out a fundamental task, that of human curation. This step allowed us to perform a quality control of the manuscripts to identify those describing validated and mechanistic models of interactions between miRNAs and protein targets. Thus, we excluded the miRNA/target associations when not validated by overexpression, silencing or genetic mutations. Finally, the manual curation allowed us to correctly standardize the gene naming, which so often diverges in the scientific literature. This final manual data standardization was necessary for the proper execution of the machine learning procedures and creation of networks. This procedure left us with a distilled set of 68 papers that we analyzed and whose results are included in this review. Cytoscape (v. 3.7.2) was used to create and visualize the networks describing the information obtained from the literature. With the aim of reporting robust findings, we start here by focusing on the miRNAs or drugs studied in at least two different scientific articles (Figure 1). 

The coding genes’ nomenclature was standardized by using the HUGO Gene Nomenclature Committee (HGNC). In the network, we used a shape code to graphically highlight miRNAs (red square), their targets (yellow circle), miRNA upstream regulators (green triangle) and each type of drug (connection) with a specific color, as indicated in the legend of Figure 1. Each connecting edge corresponds to a single publication; thus, different lines of the same color indicate a different paper. To better visualize the most connected miRNAs, the node size is proportional to its degree (the number of links between a miRNA and its targets or vice versa); we assigned to lower degrees of value a smaller size. All information, extracted during human curation, on the miRNA/target/drug and the relative references is reported in Appendix A. In the following paragraphs, we will describe the most prominent miRNA/target interactions within the context of drug resistance in colorectal cancer (CRC).

## 2. The MiR-200/MiR-181/MiR-155 CTNNB1 BCL2 Network

The members of the miR-200 family (miR-200a/b/c and miR-141) and miR-181a play a pivotal role in the multidrug resistance of colorectal carcinoma. These miRNAs were considered as suppressors of cancer growth and metastasis through the regulation of different molecular pathways. MiR-200c and miR-181a are the most-connected miRNAs participating in this network, and both inhibit catenin beta 1 (CTNNB1) expression, a key target associated with three different drug resistances (Figure 2).

The miRNAs in the network are connected with a number of targets (direct or indirect) and are involved in the resistance to vincristine (VCR), irinotecan (CPT11), 5-fluorouracil (5-FU), oxaliplatin (L-OHP), trichostatin A (TSA) and cetuximab (CET). In detail, the overexpression of miR-200c leads to the direct suppression of c-Jun N-terminal kinase 2 (*JNK2*) and indirectly to that of *JUN*, ATP-binding cassette subfamily B member 1 (*ABCB1*) and matrix metallopeptidase 9 (*MMP9*), leading, in turn, to the overexpression of TIMP metallopeptidase inhibitor 1 (*TIMP1*) and *TIMP2* in HCT8 cells treated with VCR [3]. The ABCB1 molecular transporter is also an indirect target of miR-506, a negative regulator of CTNNB1 and cyclin D1 (*CCND1*), and promotes L-OHP sensitivity in colon cancer after forced expression [4]. Juang et al. confirmed that miR-200c acted as promoter of CPT11 sensitivity in CRC cells after encapsulation in solid liposomes by suppressing the RAS/CTNNB1/ZEB pathway [5]. Consistently, the loss of miR-200 and miR-141 were related to the overexpression of the zinc finger E-box-binding homeobox 1 (*ZEB1*) and snail family transcriptional repressor 2 (*SNAI2*) (targeted by miR-200a, miR-200b and miR-141) and twist family bHLH transcription factors (*TWIST*) (targeted by miR-200c and miR-141), all contributing to the epithelial–mesenchymal transition (EMT) in 5-FU-resistant CRC [6]. Moon et al. investigated the direct correlation between the overexpression of miR-141 and the decrease of the tripartite motif containing 13 (*TRIM13*) expression in the 5-FU sensitivity of CRC and the consequent activation of apoptotic pathways [7]. Ren et al. focused their study on the antagonism between miR-141, which inhibited cancer stemness by the suppression of *CTNNB1*, and H19 lncRNA, which promoted cancer growth and L-OHP resistance acting as sponge for miR-141 [8]. Furthermore, miR-194 was reported to be ‘sponged’ by H19 lncRNA, albeit, as in most of these kinds of experiments, the stoichiometry was not reported; the restoration of the miR-194 levels led to the downregulation of sirtuin 1 (*SIRT1*), resulting in a decrease of H19/SIRT1-mediated autophagy and in an increase of 5-FU sensitivity [9].

CTNN1B, one of the most connected proteins of this network, alongside BCL2, was also targeted by miR-181 and CRNDE lncRNA. The repression of miR-181 by CRNDE determined the higher expression of CTNNB1 and transcription factor 4 (*TCF4*) miR targets with a promotion of cancer cell growth, 5-FU and L-OHP resistance in CRC cells [10]. MiR-181a also inhibited the 5-FU resistance directly targeting transcription factor 4 (*PLAG1*) and, indirectly, insulin-like growth factor 2 (*IGF2*) [11]. Furthermore, miR-181a cooperated with miR-199a and miR-30d (normally downregulated in colon cancer) to downregulate the endoplasmic reticulum chaperone heat shock protein family A (Hsp70) member 5 (*HSPA5*) and increase the TSA sensitivity in CRC cells [12]. On the other hand, miR-199a, in addition to miR-375, is one of the miRNAs that strengthen the resistance to CET. In details, miR-199a and miR-375 silenced the common target PH domain and leucine-rich repeat protein phosphatase 1 (*PHLPP1*), leading to activation of the AKT pathway and increase in CET resistance [13]. The involvement of miR-199a is the opposite for CET and TSA, since it promotes a resistance to the former (by targeting *PHLPP1* together with miR-375) while it inhibits that to the latter (by targeting *HSPA5* with miR-181a and miR-30d).

The miR-200c/*ZEB1* and miR-200c/*ABCB1* relations are confirmed in two different papers, with the first couple involved in 5-FU resistance [5,6] and the second one involved directly with CPT11 and indirectly with VCR [3,5]. Furthermore, the influence of CTNNB1 on L-OHP is confirmed by two different papers, although via different miRNAs: miR-141 or miR-181a [8,10]. Finally, *PHLPP1*, *HSPA5* and *CTNN1B* are first-order targets of several miRNA families in the context of drugs resistance. The lower and left portions of this network have genes and miRNAs that likely arise from the tumor microenvironment and are not expressed in the cancer cells themselves. MiR-204 and miR-129, acting as onco-suppressors, directly affect 5-FU resistance by targeting *BCL2*, an antiapoptotic oncoprotein, which was also downregulated by miR-204/miR-155 in L-OHP resistance. MiR-204 and miR-155 were both downregulated in tumor-associated macrophages (TAMs), due to the inhibitory role of the activated interleukin 6 (IL6)/signal transducer and activator of the transcription 3 (STAT3) pathway, with a consequent upregulation of CCAAT enhancer-binding protein beta (*CEBPB*), IL6 receptor (*IL6R*), *ABCB1* (by miR-155), *RAB22A* (by miR-204) and the shared *BCL2* target [14]. This molecular mechanism, possibly involving exosomes and validated by a coculture of TAMs and CRC cells in vitro, conferred L-OHP and 5-FU resistance to CRCs. The miR-204 activity on RAB22A, a member of the RAS oncogene family, and the promotion of chemosensitivity after miRNA’s ectopic expression was confirmed in L-OHP-resistant CRCs [15]. The resistance to 5-FU was also associated with a low expression of miR-129. After an ectopic expression of miR-129 and the consequent targeting of *BCL2*, CRC apoptosis and 5-FU sensitivity were, in fact, promoted [16]. Furthermore, miR-342 was competitively bound by SCARNA2, a non-coding RNA highly expressed in CRC tissues, thus leading to a secondary upregulation of both the epidermal growth factor receptor (EGFR) and BCL2 oncoproteins and to a sustained 5-FU resistance [17]. BCL2 is one of the most-connected proteins (together with CTNNB1) and one of the most affected by miRNA activity, as reported by a number of studies on 5-FU resistance. Nevertheless, the implications of miR-204/RAB22A on the resistance to L-OHP were reported by two independent research groups [14,15]. To understand the functional involvement of the genes in this network, we looked for the most-represented cellular pathways using Fisher’s exact test (Appendix A). The false detection rate was additionally computed to control for multiple testing [18]. As expected, RAS and PI3K are among the most-represented pathways (FDR <0.05), although the signaling by the cholecystokinin (CCK) receptor is the one spanning the most members (*n* = 7) in the network (fold enrichment of 29.9 and FDR 5.6 × 10^−7^). In both gastric and colon cancer cells transfected with the cholecystokinin 2 receptor (*CCK2R*), gastrin has been shown to enhance cyclooxygenase-2 (*COX-2*) gene expression. This key enzyme is known to play an important role in inflammation and carcinogenesis. COX-2 has been involved in hyperproliferation, transformation, invasion, and angiogenesis. In CRC, the extracellular signal-regulated kinase 1/2 (ERK1/2) and PI3-kinase pathways are also involved in gastrin-induced COX-2 expression [19].

## 3. The TP53/miR-34a Network

In the second network we describe, miR-34a and TP53 are, respectively, the miRNA and the protein node with the highest degree. MiR-34a was involved in the regulation of resistance to 5-FU and cisplatin (CDDP) (Figure 3).

The loss of miR-34a expression by CpG methylation or mutation in the *TP53* gene can determine an increase of the colony stimulating factor 1 receptor (*CSF1R*), a direct target of miR-34a and a mediator of EMT, metastasis and 5-FU in CRC [20]. CSF1R was also positively regulated by SNAIL and STAT3 levels, which negatively regulate the miR-34a. The restoration of the miR-34a levels in 5-FU-resistant CRC through the treatment with regorafenib induced the decrease of *WNT1* and, indirectly, of *MYC* and *NOTCH1* expression, leading to an inhibition of the stemness [21]. MiR-34a action was also indirectly inhibited by miR-106b and miR-17, two miRNAs that promoted both cell proliferation and CDDP resistance by silencing *TRIM8* and by the indirect regulation of MYCN signaling [22]. *MYC* and *TP53* are also two of the direct targets of miR-149 involved, respectively, in L-OHP and 5-FU action. The replacement of miR-149, normally suppressed by SNAIL2 in colon carcinoma, was associated with an inhibition of EMT and 5-FU chemo-resistance upon the targeting of *MYC* and nanog homeobox (*NANOG*) [23] and with a reduction in glucose metabolism after pyruvate dehydrogenase kinase 2 (PDK2) inhibition [24]. MiR-149 was also implicated in the L-OHP resistance regulated by a LINC00460 feedback loop in p53-mutated CRC cells (SW480/OxR), which, in turn, promoted the suppression of miR-149 and miR-150 and, thus, the overexpression of *TP53* [25]. Let-7b/f were proposed as tumor suppressor miRNAs, due to their negative regulation of the cell division cycle 34 (*CDC34*) and high mobility group AT-hook 2 (*HMGA2*) oncogenes [26]. In this article, it was demonstrated that the levels of both let-7b and let-7f were upregulated by doxorubicin (DOXO) in a wild-type p53-dependent fashion, which led to the slowing of cancer cell proliferation. The Snail-dependent upregulation of miR-146a and the silencing of the NUMB endocytic adaptor protein (*NUMB*) were associated with asymmetrical cell division in colorectal CSCs and the promotion of resistance to CET [27]. The downregulation of *NUMB* by miR-142 was also correlated with DOXO resistance in CRC cells. The miRNA-induced activation of Notch signaling determined an increase in the stemness and drug resistance [28]. It is interesting to note the bivalent position of miR-34a in two different contexts, the resistance to CDDP and 5-FU. MiR-34a can act as an inhibitor of *CSF1R*, *WNT1*, *MYC* and *NOTCH1* in 5-FU-resistant cells and promotes chemosensitivity, while it is downregulated by miR-106b and miR-17, which promote CDDP resistance. MiR-637 increased the L-OHP sensitivity by repressing *STAT3*, normally highly expressed in colon cancer. The circular RNA encoded by the homeodomain-interacting protein kinase 3 gene (circHIPK3) can compete with miR-637 in regulating cell viability, apoptosis and drug resistance [29]. *TP53* is thus inhibited by four miRNAs and interacts with different drug resistances discussed in two distinct articles [20,25]. The involvement of MYC in 5-FU resistance was reported by two different articles via different mechanisms [21,23]. The miR-149/5-FU relation was also independently validated [23,24], although, again, there was no agreement about the involved protein targets. Notch and WNT signaling are over-represented here, together with angiogenesis (FDR < 0.05) (Appendix A).

## 4. The MiR-514b and MiR128 Activities Converge on CDH1

MiR-514 and miR-128, as well as miR-340, regulate the proteins involved in CDDP, CPT11 and L-OHP resistance in colon cancer (Figure 4).

Ren et al. investigated the antagonist effects of the miR-514b-5p and miR-514b-3p products, respectively, a promoter and suppressor of metastasis, EMT and CPT11/CDDP resistance, by regulating cadherin 1 (*CDH1*) and claudin 1 (*CLDN*), the targets of miR-514b-5p, frizzled class receptor 4 (*FZD4*) and netrin 1 (*NTN1*), the targets of miR-514-5b-3p (previously shown in Figure 1) [30]. On the other hand, miR-128 was associated with L-OHP sensitivity by its indirect enhancing of *CDH1* expression and the downregulation of multidrug resistance-associated protein 5 (*MRP5*) and the *BMI1* Polycomb Ring Finger proto-oncogene. This activity was reported to also be present in the exosomes secreted by L-OHP-resistant cell lines [31]. BMI1 is a promoter of stemness traits of cancer cells and represents a key mutual target linking miR-128 and miR-340, both suppressors of tumorigenesis in CRC. In particular, miR-340 appeared to be sponged by circ_001680, leading to an upregulation of *BMI1* and to an increase of both the cancer stem cell (CSC) population and CPT11 resistance [32]. Among the key factors of this network, CDH1, an important onco-suppressor, was confirmed by two research groups. In fact, *CDH1* was downregulated by miR-514, promoting CPT11 and CDDP resistance, while it was indirectly upregulated by miR-128, which contrasted the oxaliplatin resistance. In addition, BMI was suppressed by either miR-340 or miR-128 to sensitize CRC cells, respectively, to C and to L-OHP treatments.

## 5. Smaller MiRNA Networks Involved in CRC Drug Resistance

Some smaller networks reported in Figure 1 were not discussed above, but in our opinion, they should be carefully noted. We list and discuss them briefly in the following paragraphs.

### 5.1. MiR-195

The role of miR-195 in drug resistance, depicted in Figure 1, was the object of divergent conclusions. Kim et al. sustained that miR-195-5p promotes 5-FU resistance by suppressing the WEE1 G2 checkpoint kinase (*WEE1*) and checkpoint kinase 1 (*CHK1*) in CRC [33]. Jin et al. affirmed that miR-195-5p enhanced 5-FU sensitivity and apoptosis, involving the suppression of mechanisms induced downstream by NOTCH2 and the recombination signal-binding protein for immunoglobulin kappa J region (*RBPJ*) [34]. Qu et al. concorded with the latter hypothesis of miR-195 as promoter of CRC chemosensitivity; in particular, they investigated the relation between the suppression of BCL2-like 2 (*BCL2L2*) by miR-195 and the sensitivity to DOXO [35].

### 5.2. MiR-194

This miRNA was reported to be downregulated by HMGA2 as a consequence of *VAPA* suppression by miR-194, thus leading to the sensitization of cancer cells to CPT11 and L-OHP [36].

### 5.3. MiR-15b

The overexpression of miR-15b determined the proapoptotic and antiproliferative effects and is associated with a major sensitivity to 5-FU treatment by suppressing either the Pim-1 proto-oncogene, serine/threonine kinase (*PIM1*) [37] or doublecortin-like kinase 1 (*DCLK1*) [38].

## 6. Unconfirmed Associations of MiRNAs with Drug Resistance in CRC

The three networks we discussed above were those including ‘validated’ miRNA/drug or miRNA/target interactions, i.e., those described by at least two unrelated research teams. Nonetheless, Figure 1 also contains interactions that have not been independently confirmed. We describe below these findings, albeit with a cautionary note, grouping them by drug.

### 6.1. 5-Fluorouracil Resistance

MiR-372/373 acted as promoters of stemness and 5-FU resistance in CRC cells by silencing the genes implicated in the differentiation process, such as the speckle-type BTB/POZ protein (*SPOP*), SET domain containing 7, histone lysine methyltransferase (*SETD7*) and vitamin D receptor (*VDR*) targets [39]. MiR-377 downregulated the Wnt/β-catenin pathway by targeting the X-linked inhibitor of apoptosis (*XIAP*) and *ZEB2*, with a positive effect on apoptosis and 5-FU chemosensitivity [40]. MiR-587 was considered as a 5-FU antagonist by repressing the protein phosphatase 2 scaffold subunit A beta (*PPP2R1B*) with an increased *XIAP* expression and AKT pathway activity [41]. This effect was reversed by the overexpression of *PPP2R1B* associated with a promotion of apoptosis. MiR-501 was downregulated by the KH-type splicing regulatory protein (*KHSRP*), with a consequent upregulation of its ERBB receptor feedback inhibitor 2 (*ERRFI2*) target, thus determining the 5-FU cell resistance and CRC proliferation [42]. Both effects were contrasted by either *ERRFI2* knockdown or miR-501 overexpression. MiR-199b was commonly downregulated in colon cancer, while the miR target SET nuclear proto-oncogene (*SET*) was highly expressed and correlated to 5-FU resistance in advanced rectal cancer (LARC) [43]. The ectopic expression of miR-199b determined the 5-FU sensitivity and represented a frontier to prevent drug resistance. MiR-1290 expression was highly detectable in deficient mismatch repair (dMMR) colon cancer and was associated with 5-FU resistance [44]. The silencing of miR-1290 determined an upregulation of its direct target mutS homolog 2 (*MSH2*) and a relative 5-FU sensitivity in CRC cells. Liu et al. demonstrated that LINC01296 downregulates miR-26a and indirectly upregulates the polypeptide N-acetylgalactosaminyl transferase 3 (*GALNT3*) miR target, thus promoting the PI3K/AKT pathway by the catalysis of mucin 1 (*MUC1*) and 5-FU resistance [45]. Tumor suppressor miR-22 was related to autophagy inhibition and a proapoptotic effect that led to a promoted 5-FU sensitivity [46]. From a molecular point of view, miR-22 suppressed the BTG antiproliferation factor 1 (*BTG1*) target and, indirectly, thymidylate synthetase (*TYMS*) and upregulated sequestosome 1 (*SQSTM1*), a downstream target.

### 6.2. Irinotecan Resistance

Sun et al. investigated the promoting effect of calcitriol on the miR-627 expression and demonstrated a relation between the suppression of its target, cytochrome P450 family 3 subfamily A member 4 (*CYP3A4*), and the CPT11 sensitivity in CRC cells with a relative inhibition of cell growth and an increase of apoptosis [47]. The loss of miR-4454 expression was correlated with the activation of the G protein nucleolar 3-like (GNL3L)/NFKB pathway, resulting in a resistance to CPT11 [48]. The overexpression of miR-4454 restored GNL3L silencing and reduced chemoresistance and cancer aggression in vitro.

### 6.3. Cetuximab Resistance

MiR-100 and miR-125b promoted CET resistance by suppressing the negative modulators of Wnt signaling, such as dickkopf WNT signaling pathway inhibitor (*DKK1*), *DKK3* (miR-100 targets) and APC regulator of WNT signaling pathway 2 (*APC2*), GATA-binding protein 6 (*GATA6*), ring finger protein 43 (*RNF43*) and zinc and ring finger 3 (*ZNRF3*) (miR-125b targets) [49]. MiR-302a was generally downregulated in colon cancer; its overexpression directly inhibits metastasis and CET resistance by silencing nuclear factor I B (*NFIB*) and CD44 targets [50].

### 6.4. Doxorubicin Resistance

MiR-135b acted as promoter of DOXO resistance and antiapoptotic programs by directly targeting the tumor suppressor kinase 2 (*LATS2*) [51]. These results were also confirmed in a xenograft model.

### 6.5. Oxaliplatin Resistance

*LATS2* was silenced by miR-31, itself upregulated by forkhead box C1 (*FOXC1*) in L-OHP-resistant cells [52]. MiR-107 was also a promoter of L-OHP resistance by suppressing calcium-binding protein 39 (*CAB39*) and activating the protein kinase AMP-activated (AMPK) mTOR pathway; these events could be reversed by dichloroacetate, which promoted the chemosensitivity [53]. An additional study found that high levels of miR-153, detected in 21 (out of 30) colorectal cancer patients, correlated with L-OHP resistance, as well as a sustained cellular proliferation [54]. Mir-19b acted as onco-miRNA and as a promoter of L-OHP resistance by targeting SMAD family member 4 (*SMAD4*); this link was firstly identified by bioinformatics and later confirmed in vitro [55]. MiR-203 was also correlated with the enhancement of L-OHP resistance; a high expression of miR-203 was present in three colorectal cell lines where the ATM protein kinase was its direct target [56]. MiR-21 can play a pro-metastatic role and promote L-OHP resistance in CRC cells. In fact, Bullock et al. demonstrated that an ectopic expression of miR-21 increased the invasiveness by way of an indirect upregulation of matrix metallopeptidase 2 (*MMP2*), which was, in turn, negatively regulated by the reversion-inducing cysteine-rich protein with kazal motifs (*RECK*) miR-21 target [57]. On the contrary, miR-27b, detected at low levels in L-OHP-resistant CRC cells due to c-MYC binding in the promoter of the miR-27B gene, was involved in chemosensitivity by repressing the autophagy-related 10 (*ATG10*) target, as well in the negative regulation of autophagy [58]. Rasmussen et al. investigated another key factor in the poor outcome of colon cancer patient, the downregulation of mitogen-activated protein kinase kinase 6 (*MAP2K6*) by miR-625 and the reduction of p38 signaling linked to the evasion from apoptosis and to L-OHP resistance [59]. A last miRNA involved in the promotion of L-OHP resistance was miR-122, which also activated glycolysis by an indirect upregulation of the pyruvate kinase M1/2 (*PKM2*) miR target and was proposed as a competitive ‘sponged effect’ by a circular RNA, hsa_circ_0005963 [60].

### 6.6. 5-FU and Cisplatin Resistance (Multidrug)

A lower expression of miR-223 was detected in colon cancer cells presenting mutated TP53. The ectopic expression of miR-223 in p53-mutant CRCs promoted 5-FU and CDDP sensitivity by targeting stathmin 1 (*STMN1*) and enhanced apoptosis [61]. When overexpressed, miR-497 targeted the 3’UTR site of the insulin-like growth factor 1 receptor (*IGF1R*) oncogene and determined an increase in cell death and 5-FU and CDDP sensitivity [62]. Gu et al. investigated a possible tumor suppressor role for miR-532, found to be downregulated in colorectal adenoma. Its ectopic expression determined a decrease of CRC aggressiveness in vitro and of a resistance to 5-FU and CDDP by suppressing the ETS proto-oncogene 1 transcription factor (*ETS1*)/transglutaminase 1 (*TGM1*) axis and the Wnt/β-catenin pathways [63].

### 6.7. 5-FU and L-OHP Resistance (Multidrug)

The expression of miR-4802 and miR-18a was indirectly repressed by Fusobacterium (F.) *nucleatum*, a component of the gut microbiota highly represented in drug-resistant colon cancer patients, resulting in the upregulation of autophagy-related 7 (*ATG7*) and unc-51-like autophagy activating kinase 1 (*ULK1*) targets, two activators of autophagy, as well as a resistance to 5-FU and L-OHP [64]. MiR-92a, secreted by cancer-associated fibroblasts in exosomes, was positively correlated with the tumorigenesis of colon cancer. It promoted stemness, metastasis, 5-FU and L-OHP resistance and inhibited mitochondrial apoptosis mediators, such as F-box and WD repeat domain containing 7 (*FBXW7*) and the modulator of apoptosis 1 (*MOAP1*) [65].

Finally, the non-validated interactions for drugs that have not been the object of more than one study and for this reason not included in the networks of Figure 1 are listed in Table 1.

## 7. Drug-Centric Network and Clusters of MiRNA/Targets Interactions in CRC

In this paper, we have hitherto discussed the miRNAs and their interactions, either the first or higher order, to understand the mechanisms underlying various types of chemoresistance in CRC. Protein targets were included in the network and provided the connections of non-coding RNAs with the molecular effectors in apoptosis, cell proliferation and other major cellular processes of CRC. In some of these networks, members of the other classes of non-coding RNAs, such as lncRNAs or circular RNAs, also participated. At this stage, we wished to dig further into the intricate web of gene networks by using a different point of view, namely that of an all-in drug interaction. We obtained such a view by considering the drug nodes rather than, as above, edges. The resulting network is quite complex, and we report it integrally in Appendix A, highlighting the most connected drug resistance (green rhombuses) and their relations with the miRNAs (red squares) and miRNA targets (yellow circles) in CRC. The upstream regulators of miRNAs are indicated as sky-blue triangles. The map node size was proportional to the node’s degree, and the relative statistics are listed in Appendix A. Since this drug-centric network is highly connected, unlike the one of Figure 1, we looked for embedded clusters, using a community analysis, implemented by the GLay plugin in Cytoscape. Figure 5 shows the six major clusters identified within the drug-centric network. The largest cluster, on the top left, includes the miRNAs and proteins regulating the resistance to 5-FU: miR-155, miR-342 and miR-204 are the miRNAs with the highest degrees, while BCL2 and ABCB1 are the most prominent among proteins. In the L-OHP cluster miR-92, miR-181a and miR-506 are the most connected, and CTNNB1 is the protein with the highest degree. While, in the previous two clusters, there was only one drug, CPT11 and VCR share together another cluster, with EMT gene representation (miR-200c/miR141 and ZEB1/SNAI2 and VIM). DOXO, axitinib, sorafenib and nutlin are all in another cluster, which comprises miR-17, miR-106b, let-7b/f, miR-34a and miR-146a, alongside TP53, TRIM8, MYCN and CDC34. The biological process for the seven genes in this cluster is the ‘positive regulation of cell death’ (FDR = 9.2 × 10^−3^) as calculated using the PANTHER Over-representation Test. The CET and TSA cluster includes miR-125b and miR-199a and AKT1 as a protein target. The CET/TSA cluster corresponds to Wnt signaling in the GO biological process (FDR = 1.2 × 10^−4^). CDDP spans miR-514 and miR-532, and the GO analysis points to gland development and other processes involved in cell differentiation.

## 8. Conclusions

Our data-driven and machine learning-assisted review distilled some well-defined genetic networks involved in the drug resistance of CRC. The largest miRNA network in CRC drug resistance spanned miR-200s/miR-181a, among others, and was implicated in the action of six different anticancer treatments (Figure 2). In this network, CTNNB1 plays a pivotal role, and it is at the interface of two miRNA subnetworks. CTNNB1 is part of a complex of proteins forming adherens junctions, which are important for the establishment and maintenance of epithelial cell layers by regulating cell growth and adhesion between adjacent cells [74]. *CTNNB1* is altered in 4.81% of colorectal carcinoma patients mutations, which are commonly homo- or hemizygous, indicating a higher threshold of CTNNB1 stabilization to be required for transformation in the colon as compared to extracolonic sites [75]. Moreover, different mutational hotspots in *CTNNB1* for MSI-H and MSS CRCs suggest different effects on CTNNB1 stabilization. Reduced E-cadherin may also contribute to higher levels of transcriptionally active CTNNB1, and it is not directly linked to the *CTNNB1* mutational status. Another target shared by both miR-181a and miR-200s is ABCB1, a membrane transport involved in multidrug resistance. ABCB1 links the larger portion of this network to the miR-155 lobe. MiR-155 is expressed both in CRC cells and in the tumor immune infiltrates, with the presence of CEBPB pointing to tumor-associated macrophages as additional actors in drug resistance. The potency of miR-155 indirectly regulates IL6R, which also suggests the inclusion of granulocytes in the relevant immune cells. Finally, there is a higher-order downregulation of the *BCL2* and *EGFR* oncogenes by both miR-155 and miR-342. The molecular mechanisms underlying multiple drug resistance are revealed here as crossing different types of cells and some of them appearing to be exosome-mediated.

Another network that stands out, albeit a smaller one, is highly concentrated around miR-34a [76] and comprises heavy-weight cancer genes, namely TP53 and MYC, together with some other outstanding oncoproteins, such as MYCN, NOTCH1, WNT1, CSF1R, CDC34 and the stem cell regulator NANOG (Figure 3). The notorious onco-miR-17, which is transcribed by MYC [76], seems to have an opposite influence when compared to miR-34a. This small network has been reported in the resistance to five different cancer drugs.

A small number of microRNAs and proteins in the networks and clusters that we defined through our work are critically involved in major anticancer treatments for colon cancer. In particular, the family of miR-200, miR-34a, miR-155 and miR-17 appear among the key microRNAs. Thus, the regulation of these miRNAs and their downstream targets or effectors might help to interfere with several drug resistance mechanisms in CRC. As evidenced by our study, few miRNAs seem to have pleiotropic effects on different anticancer drugs. These miRNAs and their partners might also be used in predictive hybrid coding/non-coding gene signatures to address patients to the most effective therapy.

## Figures and Tables

**Figure 1 cancers-13-04355-f001:**
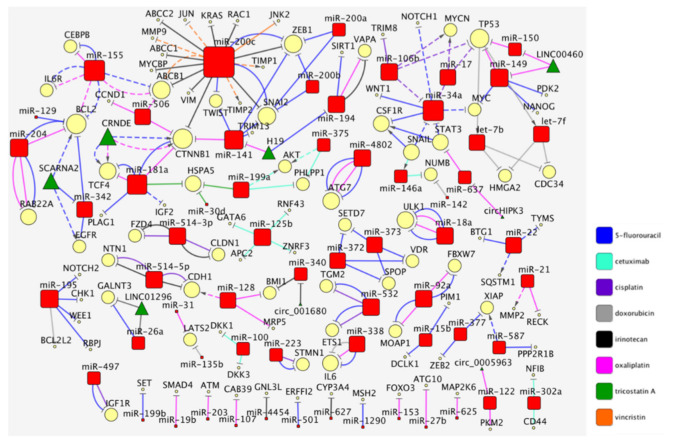
The molecular networks of miRNAs and their targets in colon cancer drug resistance. Each network shows the miRNAs/targets (nodes), or drug resistances (edges) described in at least two articles. MiRNAs are identified with red, rounded squares and the targets with yellow circles. The connecting edges corresponds to the drug resistance (color-coding for the drugs is reported in the legend). We used continuous lines for pairwise (first order) interactions and dashed for secondary (higher order) ones. Flat arrows indicate repression, while pointed arrowheads indicate activation. The map size of the miRNAs (red squares), targets (yellow circles) and non-coding RNA upstream regulators (green triangle) depends on the node degree. The data used to generate the networks are listed in Appendix A.

**Figure 2 cancers-13-04355-f002:**
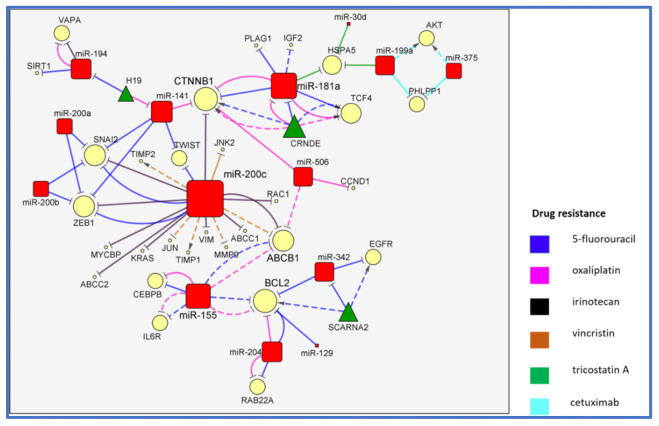
MiR-200s/miR-181a and their targets in CRC drug resistance.

**Figure 3 cancers-13-04355-f003:**
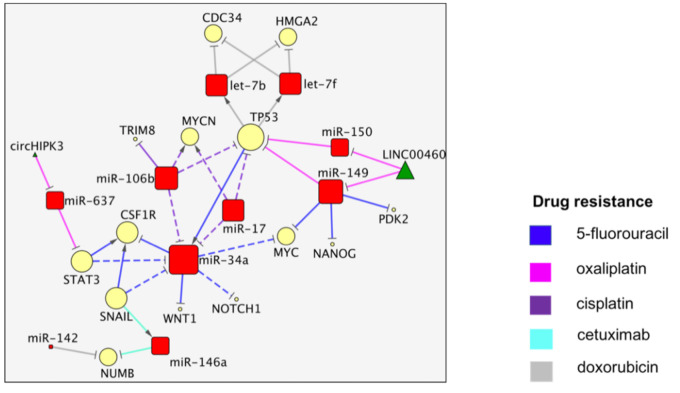
The miR-34a/TP53 network.

**Figure 4 cancers-13-04355-f004:**
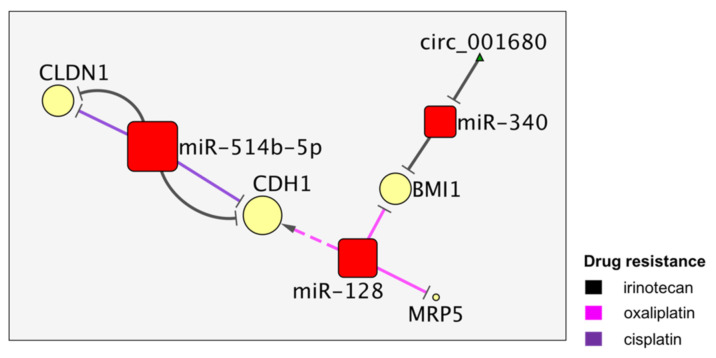
The miR-514b and miR128 microRNA niches are connected by CDH1.

**Figure 5 cancers-13-04355-f005:**
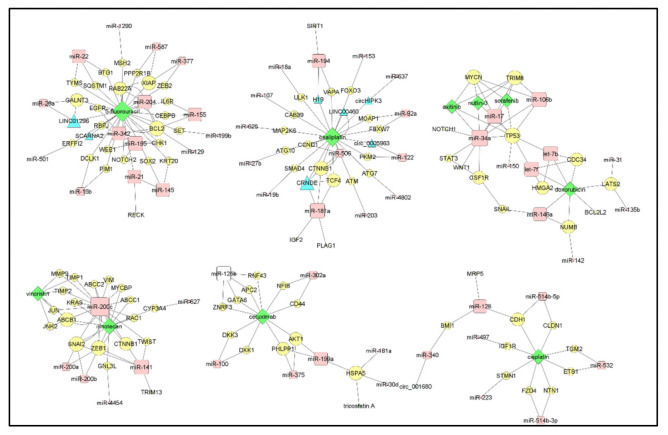
Clusters of miRNAs/targets/upstream regulators connected to the most-studied drugs in the treatment of CRC. Each subnetwork represents a separate cluster of the major drug-centric network (Appendix A). We included miRNAs (red square), their targets (yellow circle) and miRNA and target upstream regulators (sky-blue triangle) connected to the most-studied drug resistances (green rhombus). The map node size was dependent on the nodes’ degree. To build the network, we arbitrarily linked the protein targets or the ncRNA regulator with the drug and the miRNAs to either their targets or ncRNA regulator. The edges here are undirected and, thus, represent associations. Drug abbreviations: 5-FU, 5-fluorouracil; L-OHP, oxaliplatin; CPT11, irinotecan; CET, cetuximab, CDDP: cisplatin.

**Table 1 cancers-13-04355-t001:** List of miRNA target interactions and relative drugs not included in the Figure 1 networks.

PMID	miRNA	Target	Drug Name	Ref.
29844307	miR-550a	YAP1	vemurafenib	[66]
28327152	miR-106b, miR-17	miR-34a, MYCN, TP53, TRIM8	sorafenib, nutlin-3, axitinib	[22]
33585440	miR-214	KPNA3	mitomycin	[67]
28069878	miR-218	MALAT1	FOLFOX	[68]
30831320	miR-192, miR-215	NID1	doxicyclin	[69]
31208913	miR-338	IL6	cyclophosphamide	[70]
28189050	miR-675	VDR	calcitriol	[71]
30103475	miR-324	SOD2	4-acetylantroquinonol B	[72]
25928322	miR-145, miR-21	NUMB, CD44, KRT20, SOX2	5-FU and L-OHP mix	[73]

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
