# Peer review of "The Molecular Networks of microRNAs and Their Targets in the Drug Resistance of Colon Carcinoma"

_cancers, 2021, doi:10.3390/cancers13174355_

Round 1
Reviewer 1 Report
On the whole the Authors did a great effort in the aim to summarize and link existing reports to create a network of miRNA sand their targets relevant for drug resistance/sensitivity in colon cancer. However, in certain points they are too scanty in giving info about a relevant target and the link between mi-RNA-mediated expression/repression of the target and the resistant/sensitive phenotype is missing. When they mention any given target they should be more accurate and describe what the target is and what it does. The review should be re-organized in such a way that many results should be explained in more detail, and possibly discussed, along the manuscript, at the moment they are described. Otherwise it's really difficult for the reader to approach the paper.
Specific points:
Simple Summary: We reviewed the recent scientific publications describing the role of microRNAs on the regulation of drug resistance in colon cancer. T and clarify the intricate web of resulting genetic and biochemical interactions we used a machine learning approach and visualized it as a network.
- Something is obviously missing in this sentence
- Legend of Fig.1 Please add the color coding for the drugs and do not refer to Figure S1. The reading must be self-sustaining and independent of the supplementary Figures
Lines 60-64: In the network, we graphically highlighted miRNAs with a red rounded rectangle, their targets with a yellow circle, miRNA upstream regulator with a green triangle and each type of drug (connection) with a specific colour showed in legend (Figure S1). We used a contiguous line with a flat arrowhead (for an inhibitory relation) to identify a first-order target of each miRNA and dashed line (with a flat or traditional head) to a secondary (higher order) relation.
- This is the exact repetition of what already described in the legend of Figure , it is redundant.
Lines 82-84: The miRNAs in the network are connected with a number of targets (direct or indirect) and are involved in the resistance to vincristine (VCR, orange), irinotecan (CPT11, black), 5-fluorouracil (5-FU, blue), oxaliplatin (L-OHP, fuchsia), tricostatin A (TSA, green) and cetuximab (CET, sky-blue).
- Info about the meaning of the different colors should be indicated in the legend not here
Lines 101-102 albeit, as in most of these kinds of experiment the stoichiometry was not reported
- albeit, as in most of these kinds of experiment, the stoichiometry was not reported
Lines 119-120…. 5-FU resistance [5],[6] and the second one involved directly with CPT11 and indirectly with VCR [3],[5].
- refs should be cited sequentially in the same bracket [5, 6]… [3, 5]
Line 146: … As expected, Ras and PI3K are among….
- Ras should be capitalized (RAS)
Line 158:…. miR-34a was involved in the regulation of resistance to 5-FU (blue lines) and cisplatin (purple lines) (Figure 3).
Line 169: targets of miR-149 involved respectively in L-OHP (fuchsia lines) and 5-FU action.
Lines 198-200: miR-514, miR-128, as well as miR-340 regulate proteins involved in cisplatin (purple lines), irinotecan (black lines) and L-OHP resistance (fuchsia lines) in colon cancer (Figure 4).
Lines 334-336: The resulting network is quite complex, and we report it integrally in Figure S2, highlighting the most connected drug 335 resistance (green diamonds), and their relations with miRNAs (red rectangle) and miRNA 336 targets (yellow circle) in CRC.
- Info about the meaning of the different colors should be indicated in the legend of the figure not in the text
- In general the Authors use the acronym to indicate a gene targeted by a certain miRNA: however, even though “famous” genes (like for example, MYC, RAS, TP53, etc) don’t need an explanation many genes they mention (such PHLPP1, TRIM8, RAB22A, PDK2, CDH1, CLDN, FZD4, NTN1, MRP5, RBPJ, BCL2L2, VAPA, DLCK1, SPOP, STD2, VDR, PPP2R1B, KHSRP, ERFFI2, GALNT3, BTG1, SQSTM1, GNL3L, RNF43, ZNRF3, NFIB, LATS2, FOXC1, CAB39, RECK, PKM2, STMN1, ETS1, TGM1, MOAP1, CEBPB) as are not so familiar for a general audience. Therefore, a line of explanation, or at least the indication of the full name would be helpful for the reader. Otherwise, the reader will be simply confronted by a “phone book-like” list of miRNA and their target associated to resistance or sensitivity to a drug without having a hint of a possible role in the biological process examined. I recommend the Authors to be more exhaustive in defining and pointing out links between mi-RNA/target and target/biological role
Line 174: in p53-mutated (SW480/OxR), which in turn promoted
- in p53-mutated CRC cells (SW480/OxR),…
Line 180: …Snail-dependent upregulation…
- Snail should be capitalized (SNAIL)
Line 183: …activation of Notch signaling….
- Notch should be capitalized (NOTCH)
Line 185: …resistance to CDDP…
- What drug is this? Please introduce it before using an acronym! Even though a list of abbreviation is indicated as the final paragraph of the manuscript, abbreviations and acronyms should be explained at the moment of the first use in the text
Line 221: MiR-195
Line 228: MiR-194
Line 221: MiR-15b
- Since each miRNA is the subject of a small discussion they should in some way emphasized, like by using an italic character and making both of them the start of a new line
Line 221: why abbreviate dichloroacetate (DCA) if it is not used anymore in the manuscript, but not give the full name of all those “mysterious” genes listed throughout the paper?
Line 304: Lower expression of miR-223 were detected
- Lower expression of miR-223 was detected
Line 371:… beta-catenin (CTNNB1)…
- Why the authors abbreviate beta catenin at the end of the paper after having used the acronym wherever else? In addition sometimes they indicate b-catenin, some other beta catenin, some other CTNNB1… very confusing! If it the acronym that they want to use just write it in extenso the first time putting the acronym in brackets and then be consistent throughout the paper
Reviewer 2 Report
This is a review of molecular, miRNA related networks in colorectal cancer. The field is highly complex and the effort to clarify the issue and provide a wider picture is commendable.
I have a few specific comments:
- Some introductory paragraph is needed as the paper is very condensed and therefore fairly difficult to read and understand.
- Style and language improvement is needed, for example: "which transcription was reported to be driven by MYC" needs correction in line 395, MiR-195 is "hanging" in line 220 and MiR-194 in line 228. .
